# Verification of Psychophysiological Effects of Satoyama Activities on Older Adult Volunteers and Young People in Post-COVID-19 Society: A Case Study of Matsudo City, Japan

**DOI:** 10.3390/ijerph191710760

**Published:** 2022-08-29

**Authors:** Qiongying Xiang, Zhengwei Yuan, Katsunori Furuya, Takahide Kagawa

**Affiliations:** Graduate School of Horticulture, Chiba University, Chiba 271-8510, Japan

**Keywords:** psychophysiological effects, Satoyama activity, UGS, psychological restoration, post-COVID-19 society

## Abstract

Many scholars have focused on Satoyama, which is characterized by mountains or villages away from the urban spaces. Our objective is to verify its psychophysiological effects on people performing usual Satoyama activities in ignored, small urban green spaces to help people find ways to stay healthy in post-coronavirus disease 2019 (COVID-19) society. In this study, 12 older adult volunteers from the “Forest around the Mountains” Nonprofit Organization and 12 young people from the university were invited as study subjects. They were asked to observe nature for 10 min and work for 30 min in the small green space “Forest around the Mountains”. The Profile of Mood States 2nd Edition (POMS) and the State-Trait Anxiety Inventor (STAI) were used as psychological scales to detect their psychological restoration and blood pressure before and after the Satoyama activity. Their heart rate during the activity was used as the physiological indicator. The study showed that, as Satoyama volunteers, the older adults group had significant restorative psychophysiological effects during this experiment compared to the younger group; their systolic and diastolic blood pressure dropped significantly after Satoyama activities, On the other hand, the young group have been in the normotensive range before or after Satoyama activities The psychological indicators such as Anger-hostility, Confusion-bewilderment, and Tension-anxiety were significantly lower in the younger group but were still significantly higher than the indicators of the older adults. In general, this study found that Satoyama activities benefited young and older participants, especially older adults with high blood pressure. Satoyama activities in small urban green spaces are thus necessary and worth promoting in the post-COVID-19 era.

## 1. Introduction

### 1.1. Background of the Post-COVID-19 Society

Since the second industrial revolution, the high density of people living in cities and urban developments has become the norm [1]. However, such high density increases the spread of infectious diseases, such as the Spanish pandemic in 1919 [2,3,4] and the recent coronavirus disease 2019 (COVID-19) [5]. Since 2019, the governments of many countries have taken various measures to face the pandemic [6,7,8], for example, by introducing home quarantine policies [6], restrictions on going outside unless necessary [7], and implementing “no confined space, no dense places, no close contact” in Japan [8]. As proven by researchers, governments’ policies of restricting outings and people’s fear of viruses may have caused some health problems during this period, including adverse psychological effects, such as post-traumatic stress symptoms, confusion, anger, fear and discrimination, anxiety, depression, and stress regarding the sustainability of their current jobs [9,10,11,12]. This socially destructive stress can cause serious problems related to the proper functioning of the immune system [13]. Furthermore, the segregation policy in the post-COVID-19 era may cause some people, especially younger ones, to suffer from psycho-logical illnesses owing to a lack of socialization [6]. These problems also exist in Japanese society: the data from 2019 to 2021 show that young people living in Japanese cities during the post-COVID-19 era are experiencing a devastating effect on their studies and possibly their health [14,15,16]. Moreover, as Japan is an aging society, older adults can easily become severely impacted patients due to aging and diseases [17].

### 1.2. Satoyama in Small Urban Green Space (UGS)

In the post-COVID-19 era, many kinds of urban green space (UGS) are still proving to be beneficial to people’s physical and psychological health [18,19,20,21]. While the opening of urban parks or gardens has become erratic at times [22], owing to restrictions on capacity, many small areas of public green spaces have continued to offer a relaxing natural environment, although they are often neglected [23,24]. Natural areas between cities and villages, such as grasslands, rice fields, drylands, and secondary forests, are widely de-fined as Satoyama [25,26]. In Japan, the rapid urban development in the 1970s forced some secondary forests used as fuel charcoal on the outskirts of cities to be left behind by the expanding cities [27]. By the 1990s, the concept of Satoyama was expanded, and these small green spaces could be used for new types of Satoyama activities: forest therapy [28], environmental education [29,30], and concerts, in addition to forestry and agriculture [31]. Many scholars have focused on Satoyama in terms of management [32], citizens’ aware-ness [33], and environmental education [34], which is provided in the mountains or villages away from crowds. Several studies have also shown that walking and sitting during forest therapy in green spaces can have significant psychologically restorative effects on people [35,36,37,38,39,40,41]. Furthermore, Uehara has shown that Satoyama activities in different sea-sons can have a significant psychological restorative effect on people [30].

However, almost all the work reported above was performed under psychological conditions manually controlled by experts, both in the laboratory and in the field, and the psychophysiological effects of conducting Satoyama activities among people of a wide age range—especially those aged 65 years and over and young people under the age of 30—are still unknown. Most of the volunteers who have participated in Satoyama activities in Japan are older adults, and they have also been participating in them during the post-COVID-19 era.

### 1.3. Purpose

There are some questions and assumptions, which are posed as follows: (1) Do small green spaces in cities have restorative effects on people? It is assumed that there is a re-storative effect in green spaces. (2) Do Satoyama activities have restorative effects for different groups, especially older people who regularly volunteer in the forest and young people in school? It is assumed that it works better for young people who have no experience of Satoyama. (3) Is it appropriate for people to conduct Satoyama activities during the COVID-19 pandemic, and do the experiments conducted in this period have any reference value for the recovery of people’s physical and psychological health?

## 2. Materials and Methods

### 2.1. Study Site

As shown by the map in Figure 1, the “Forest around the Mountain” is in the area between Gogo and Kanegasaku, Matsudo City, and the Chiba Prefecture. Its latitude is 35.806886 and its longitude is 139.951435. The total area extends for about 2 hectares, including 200 m from north to south and 100 m from east to west [42]. The volunteers who maintain the forest are the students who attended the Satoyama Volunteer Orientation Course held in Matsudo City in 2004. Since June 2005, they have been performing conservation activities in the forest, such as cleaning, mowing, and conducting ecological surveys (Figure 2). This forest is surrounded by fields and has been called the “Forest around the Mountain” since then. In the past, the area also included spring water ponds, and the forest was mainly used to supply charcoal and compost. Later, the ground was covered with broad-leaf bamboo [43], Quercus serrata [44], and Setaria viridis [45]. The ecosystem is rich, with chickadees [46], Japanese bush warbler [47], and owls [48]. The volunteers’ future goal is to preserve the valuable natural environment left in the city and to turn it into a forest that is loved by the citizens.

### 2.2. Study Subject

To ensure that the experiment would not be affected by insects in the forest and the muggy environment, the participants—both older and young people—agreed to participate between March 2021 and April 2021. The study participants included those with no heart disease, who were in good health, and could work for about 30 min without any problems. Older adults were aged 65+ years; volunteers were from the “Forest around the Mountain” Nonprofit Organization (NPO) [49] of Matsudo City, and young people were mainly university and graduate students insured by the Health Mutual Aid Association. Recruitment was started from January to March 2021. The details of recruitment included research purpose, description of experimental content, physical condition requirements of participants and contact information of researchers. The recruitment method for older adults is to distribute flyers among them during Satoyama regular meetings, while using the Social Network Services to recruit young people. As the study entailed handling hu-mans’ private information, we required ethical approval from the Institutional Review Board of the Psychological Ethics Committee, approved by the Institutional Review Board of the Psychological Ethics Committee in November (protocol code 20-05). The experiment was recognized from November 2021 to October 2022.

### 2.3. Experimental Design

Six people participated for each experiment, and a total of four experiments were conducted: two experiments for the older adult volunteers’ group in March (*n* = 12) and two experiments for the young people group (*n* = 12) in April. As shown in Table 1, (1) after arriving at the “Forest around the Mountains,” participants gathered in a “working square” that is shown as the map in Figure 1; (2) we first measured the human body temperature with an infrared thermometer; (3) staff explained the flow and the approvement from the Human Research Ethics Review Board of the experiment and participant signed consent; (4) before starting the activity, blood pressure, Profile of Mood States 2nd Edition (POMS) [50], and State-Trait Anxiety Inventory (STAI) [51] then, participants were connected to a heart rate monitor; (5) participants took 10 min to observe nature from the entrance to the garden and fields, as shown in Figure 1; (6) after observing nature, participants took 30 min to work on the “playing square” (Figure 1), and their Satoyama activities included weeding, pruning branches, and simple logging (Figure 3); (7) blood pressure, POMS, and STAI while performing Satoyama activities were measured, and heart rate was measured continuously on nature observation and Satoyama work; (8) The heart rate measuring device was removed and finishing the experiment. Because Satoyama is different from other natural landscapes, the NPO group of “Forest around the Mountain” headed by Mr. Iki also provided helmets, mosquito repel-lent incense, sickles, and saws for this experiment.

This study was based on Catharine Ward Thompson’s unsupervised natural expo-sure method to measure the association between salivary cortisol, stress, and green space [52]. We also differentiated forest bathing [35,36,37,38,39,40,41] from harsh human-controlled conditions by simulating a working environment and conditions almost identical to those in which Satoyama volunteers usually work, including nature observation and forestry operations, which are common in Satoyama activities [31]. The forest work consisted of cut-ting, carrying branches, and weeding, which is usually a more routine activity for the volunteers, taking place twice a month. The volunteers followed the guidance of Mr. Iki, who started working in Satoyama in 2005 and did not stop during the COVID-19 pandemic; they were able to experience and observe Satoyama while working under a condition of familiarity with the status of Satoyama. Since some young participants did not belong to NPO, we separated observation and work to make the experiments psychological conditions fairer. The 30 min of work were considered on the premise that this time is recognized as the most beneficial to the body [53]. Participants were asked to pay attention to some factors the day before the experiment, including eating breakfast; sleeping early; avoiding alcohol, tobacco, and rehabilitation drugs at least 24 h before and during the experiment; and avoiding intense activities or caffeine at least 12 h before the experiment. They were not to eat any food or drink liquid (except water) for at least 1 h before the experiment [35,36].

### 2.4. Experiment’s Psychological Equipment and Questionnaire

Under the Japanese policy against COVID-19, infrared remote thermometers (TOA NUTRISTICK company) were required. If the subject’s temperature was higher than 37.5, they could not participate in the experiment. An LCD, digital, wireless indoor/outdoor thermometer and hygrometer measured the temperature and humidity of the “Forest around the Mountain.” During the experiment, systolic and diastolic blood pressures were measured before and after the experiment using a digital automatic blood pressure monitor (Omron HEM-6161), which is a battery-operated wrist measurement with the convenience of portability and measurement. It is worn on the wrist so that the hand is at heart level, and blood pressure data are measured. Heart rate and blood pressure were measured as physiological parameters; heart rate was measured continuously using a heart rate blue sensor (My Beat WHS-3). An iPad was also needed to connect to the heart rate blue sensor, which captured the signal and displayed the image.

The Profile of Mood States 2nd Edition (POMS 2) and a new short version of the STAI in Japanese were used as psychological parameters in this experiment. POMS 2 is a reliable and effective psychological response measurement, which contains 35 questions and conforms to six emotional states. We express these emotions in abbreviated states: “anger-hostility” is abbreviated as “A-H”; “confusion-bewilderment” is abbreviated as “C-B”; “depression-dejection is abbreviated” as “D-D”; “fatigue-inertia” is abbreviated as “F-I”; “Tension-anxiety” is abbreviated as “T-A”; and “vigor-activity” is abbreviated as “V-A.” Each item was measured on a 5-point Likert scale ranging from 0 (none at all) to 4 (extreme) to assess the participants’ emotional states. Except for the positive correlation between the score of V-A and people’s positive emotional state, the rest were negatively correlated. The state of anxiety part of STAI was used to measure the current anxiety state of individuals, and it contained 20 questions. A 4-point Likert scale ranging from 1 (not at all) to 4 (very much) was used to assess the participants’ anxiety status. Because the younger group included a number of Chinese people, we also adopted simplified Chinese versions of POMS and STAI, and their content was no different from that of the Japanese version [54,55].

### 2.5. Analysis Methods

The blood pressure data used in the experiment were divided between older volunteers and younger people and before and after the Satoyama activity. For mean heart rate values, we considered the mean heart rate values per minute during nature observation and Satoyama work as well as the mean heart rate for each group of younger and older volunteers for analysis. Since the sample size is less than 13 (*n* = 12) and does not satisfy the normal distribution, the Wilcoxon signed-rank test [56] was used to assess the differences in mean psychophysiological values between the two groups before and after the Satoyama activity.

## 3. Results

### 3.1. Environment, Participants, and Satoyama Activity Information

The experiment was measured using an LCD, digital, wireless, indoor/outdoor thermometer and hygrometer and was completed over 4 days: 23 and 29 March and 24 and 28 April. The environmental physical indicators were measured in three times per experiment, we used the average values as results. The temperatures on these 4 days were 15.2 ± 2.66 °C, 23.5 ± 2.34 °C, 20.825 ± 1.08 °C, and 22 ± 0.88 °C, respectively. The relative humidity was as follows: 54.25 ± 8.22 %rh, 75.25 ± 6.95 %rh, 39 ± 0.82 %rh, and 51.25 ± 1.71 %rh. The luminosity in Klux was 40.03 ± 10.60, 26.77 ± 12.68, 28.75 ± 22.27, and 24.67 ± 13.87. Sound in dB(A) was 40.9 ± 1.3, 44.5 ± 3.12, 41 ± 1.41, and 47.33 ± 5.86 (Table 2).

The average age of the older adult volunteer group was 72 ± 5.08, which was consistent with the definition of the older adults (age ≥ 65), and that of the young group was 25.4 ± 1.56. Among them, p-value represented the significant difference in each indicator between the two groups of older adult volunteers and young people. Of the older adult volunteers, 58.30% were men and 41.70% were women. The proportion of men and women in the group of young people was half and half, without significant difference (*p* = 0.755). While 83.30% of the people in the older adults’ group were unemployed, all young people (school students) had no formal work employment (*p* = 0.514), showing no significant difference. The final educational qualifications of the older adult volunteers were as follows: 25% had received high school education; 58.30% had attended university, and 16.70% had attended graduate school. The final education of young people was as follows: 75% had attained university education and 25% had attended graduate school (*p* = 0.671), which shows no significant difference. Income of per month was measured in Japanese yen (JPY). For the older adult group, 8.30% had an income of less than 100,000 JPY; 41.70% had an income of 100,000 to 200,000 JPY; 41.70% had an income of 200,000 to 300,000 JPY, and 8.30% had an income of more than 300,000 JPY; conversely, 91.70% of young people earned less than 100,000 JPY and 8.30% earned between 100,000 and 200,000 JPY. The p-value was under 0.01, showing a significant difference. In terms of smoking habits, the two groups were completely consistent, with 8.30% of smokers and 94.70% of non-smokers (*p* = 1), showing no significant difference. Moreover, 58.30% of the older adult had drinking habits, compared to 25% of young people (*p* = 0.178); finally, 41.70% of the older adult and 33.30% of young people slept between 5 and 7 h, with 58.30% and 66.70% in the range of 7 to 9 h (*p* = 0.755), respectively (Figure 4).

Figure 5 and Figure 6 analyze the participants’ usual outdoor activities and sports habits. The use of UGS includes Satoyama activities, as well as running, cycling and walking in green public facilities. Sport exercise includes not only outdoor sports but also indoor sports. It was found that the average number of times the elderly group participates in Satoyama activities per month is 2.92 and for the young people the number is 0.25. There is little difference between the two in the use of UGS, the average for a young person is 4.33 and for an old person it is 4.67. However, in terms of sporting activities, the number of 10.92 times per month for the elderly is much higher than that of 5.58 times for the younger participants, demonstrating significant differences *(p* ≤ 0.01). In the understanding of Satoyama knowledge, the Likert scale was used for testing, and it was found that the value was in the middle, thus the elderly group did not belong to the type what they did not know at all or knew Satoyama completely. The young people’s group is distributed across all levels of understanding of Satoyama: 25% do not know Satoyama at all and 8.3% know Satoyama very well. This is in line with Japan’s society, which has led to many elderly people being active in various NPO groups after retirement, whereas the involvement of the younger group is rare. According to a survey conducted by Ichihara et al., the average age of volunteer groups in Japan is 40% over 60 years old and 40% between the ages of 50 and 60 [57,58]. On the other hand, the impact of multimedia, childlessness, underpopulation, and the transition to homeopathy has reduced communication between urban and rural areas, causing young people to move away from nature [28]. 

### 3.2. Reliability Analysis of Physiological and Psychological Subsection

The parameters of participants in Satoyama activities (before and after working) are provided in Table 3. The overall parameters show internal consistency (0.5–0.9). The results indicate that heart rate and blood pressure, as physiological parameters, had good internal consistency (more than 0.7) except the heart rate of young people, while POMS and STAI scores had good internal consistency (more than 0.7). The alpha reliability of heart rate of older adult volunteers was 0.511 with internal consistency, while it of young people was −1.660 without internal consistency. For blood pressure, it was 0.684 and 0.863, respectively, for the two groups. Cronbach’s α reliability analysis between the older adult volunteers (*n* = 12) and young people (*n* = 12) showed that the reliability of the POMS score for the two groups was 0.767 and 0.795 with good internal consistency, respectively; and that of the STAI score was 0.806 and 0.680 with good internal consistency, respectively [59] (Table 3).

### 3.3. Physiological Effects

Table 4 compares the physiological data between the two groups of older adult volunteers and young people. We used the Wilcoxon signed-rank test for the testing of *p*-values. The data were shown as mean and standard deviation [60].

The average value of systolic blood pressure for the older adults before activity was found to be 153.33 ± 26.46 and that of young people was 114.58 ± 11.603 (*p* < 0.05). A significant difference was observed between the two groups. After the activity, the value of the older adults’ group was 137.92 ± 16.88, and that of the young group was 113 ± 12.068 (*p* < 0.05). With regard to diastolic blood pressure, the average value of the older adults before activity was 91.58 ± 10.92 and that of young people was 74.25 ± 13.66 (*p* < 0.05). A significant difference was observed between the two groups and in the blood pressure of the older adults before and after the Satoyama activity; after the activity, this value was 84.08 ± 12.75 for the older adults and 73.75 ± 14.04 for young people. No significant difference was found between the two groups. Comparing the heart rate between nature observation (1–10 min) and work in Satoyama (11–41 min), a significant difference was observed in the former between the older adults and young people, while no significant difference was found in the latter (*p* < 0.05), and the heartbeat value of young people was higher than that of the older adults.

### 3.4. Psychological Effects

The psychological data of POMS and STAI between the two groups of older adult volunteers and young people were compared by using the Wilcoxon signed-rank test (Table 5). With regard to POMS, the negative psychological benefits of POMS before Satoyama activity were as follows: the A-H of the older adult volunteers’ group was 2.17 ± 2.04 and that of the young group was 5.583 ± 3.23 (*p* < 0.05); the C-B was 1.75 ± 2.09 for the older adults’ group and 8.17 ± 4.09 for the young group (*p* < 0.01); the D-D was 1.67 ± 1.61 for the older adults’ group and 7 ± 4.07 for the young group (*p* < 0.01); the F-I was 2.75 ± 3.60 for the older adults’ group and 10.5 ± 5.05 for the young group (*p* < 0.01); finally, the T-A was 1.92 ± 1.73 for the older adult group and 7.83 ± 3.74 for the young group (*p* < 0.01). The results show that the value of negative emotions of the older adults’ group is lower than that of the young group, and the difference between the two groups is significant. No significant difference was found between the two in V-A (*p* = 0.272), an option to express positive emotions Regarding the negative psychological values of POMS after Satoyama activity, the A-H, C-B, D-D, and T-A equivalents of the young group decreased, and significant difference was still observed with the older adults’ group. The p-value of C-B was 0.004 and the F-I was 0.003 (*p* < 0.01). The two values of the young group were higher than those of the older adults’ group, with significant differences. The values of STAI de-creased after activity between the two groups, and no significant difference was observed. In addition, after using the Wilcoxon signed-rank test to check the values of the older adults’ group, we found that A-H, T-A, V-A, and the STAI score changed significantly (*p* ≤ 0.05), while A-H, C-B, D-D, and T-A, was found significant changes in the young group (*p* ≤ 0.05).

## 4. Discussion

In the past, when the Japanese government planned a city, the green space belonging to secondary forests was still in the suburbs, and the area was used for forestry to provide charcoal. However, with the expansion of cities [27], the suburbs gradually became cities, and secondary forests were slowly abandoned due to the use of new energy sources, such as oil and gas. During the COVID-19 pandemic, many closure policies led to a reduction in the use of large public green spaces in the city, although people needed nature and outdoor activities to keep fit. Therefore, this study addresses the use of small green spaces for people’s health in the city.

It is worthwhile to look ahead to the new use and development of these spaces as Satoyama. Unlike walking and observation in experiments with general restorative activities conducted in the forest, this study utilizes the Satoyama activity, a popular form of mountain forest management in Japan. Satoyama activity is a cultural service and an eco-logical service, and it is worthy of being promoted all over the world as “forest bathing”.

Von Lindern [61] has shown that occupational involvement is a constraint to the recovery of leisure time in forest environments. However, Table 4 shows that although the physiological vulnerability of older adults can be seen in their hypertensive at the outset, systolic blood pressure had a mean value of 153.33 ± 26.46 before the Satoyama activity, it had decreased to 137.92 ± 16.88, after a period of nature observation and work in the Sato-yama; diastolic blood pressure also decreased from 91.58 ± 10.92 to 84.08 ± 12.75, near the normal level [62]. Satoyama activities have an important role in the physical health of older adult volunteers. On the other hand, young people have been in the normotensive range before or after Satoyama activities, which have more impact on their heart rate (Table 4). High heart rate in young people has been related to their high basal metabolism [63,64], however, the accuracy of the data is limited so we can only describe the heart rate inside the limitation. 

After comparing the physiological and psychological data between the two groups separately, Satoyama activity has a significant effect on both older adult volunteers and young people. For the older adults, their systolic and diastolic blood pressure dropped significantly after Satoyama activities, which means Satoyama activities were found to be effective for the older adults with high blood pressure. A-H, T-A, and STAI, which indicate negative emotions, showed a significant decrease; V-A showed a significant increase, which indicates that the frequency of participating in Satoyama as a volunteer about three times a month benefits the body (Table 5).

For young people, their physical health did not change significantly before and after the Satoyama activities; the psychological indicators such as A-H, C-B, D-D, and T-A showed a significant decrease, and the anxiety shown by STAI had not changed significantly (Table 5). Compared with the older adults’ group, after Satoyama activities, A-H, C-B, F-I, and T-A were higher than the data of the older adults’ group with a significant difference. It is also reflected in the fact that young people had too little exposure to Satoyama, also the negative psychological state is usually prominent given their lower income, bad lifestyle habits, and the exertion and confusion of the Satoyama activity process were much greater than that of older adults, who were frequently active in Satoyama. Indicating that Satoyama activities have better restorative effects for older adult volunteers, for young people who live in cities, Satoyama activities also have an essential restorative effect on their psychological health (Table 5).

## 5. Limitations

This study has some limitations. In post-COVID-19 pandemic conditions, everyone has to breathe through a mask, and people cannot fully enjoy the natural air of the forest, increasing the level of hard work. Moreover, individuals do not work in the same way, and even if they are scheduled for the same work, their physiological indicators—especially heart rate indicators—will differ. According to a journal on exercise, heart rate changes from a rapid increase to a slow decrease after a period of exercise [65]. Thus, as Satoyama activities in our experiment were not highly difficult jobs but gardening activities, such as weeding, pruning branches, and simple logging, we could not determine changes in heart rhythm every minute have meanings. In addition, nature observation in the mountains does not entail merely sitting in a certain position but walking from the entrance to the end of the forest. Therefore, from the first 10 min of activity to when the participants faced the work, a rise in heart rate, but also a gradual leveling off, occurred [65]. This was the first time the young people faced Satoyama, and the Fatigue-inertia (F-I) of the POMS and STAI test showed that they were more tired and anxious (Table 5). Because of their high basal metabolism, sweating during the activity also affected the effectiveness of the sensors attached to the skin resulting in unsatisfactory heart rate data for the young people [64,65,66]. This makes the heart rate data not distinguishable between observation and work; if enough people were included, it may be possible to determine whether the physiological changes are consistent across the different types of participants in the Satoyama.

A comparison of the physiological measurements of young people and older adults people before and after the activity was performed, but because the sample size was too small and although there was a recovery effect was observed, no significant change in the younger group was found, so it was discarded as a description of the findings, which excluded it from the results.

The participants were asked to answer a questionnaire about the future development of Satoyama in the pandemic, which was written in free form. Although the sample was too small to be reflected as a result, the questionnaire revealed many valuable comments, as follows:

With regard to coronaviruses: “The Satoyama of the future has a large area, fresh air, and distribution. This predominance has a marked advantage in the era of new corona-viruses”; “As an outdoor natural space, the new coronavirus has little effect”; “The Sato-yama is very beautiful and will surely attract many people after the coronavirus. I was very happy with this experience. I would like to come back if I have the chance. Staff working hours can be reduced to an acceptable level to prevent the spread of the pan-demic.”

With regard to the home: “I hope more and more people realize that while it is nice to go out in the city, it is also nice to go out in the woods, in nature, which is closer to home.”; “And amid the COVID-19 pandemic, when people stay at home for a long time, they inevitably want to go outside for activities.”

With regard to Satoyama: “The Satoyama is a spacious environment that avoids be-ing dense and is beneficial even with the COVID-19 disaster”; “I believe that protecting and, if possible, expanding Satoyama will improve the health of many people”; “We will prevent many people from entering the Satoyama as well”; “Although we do not believe that COVID-19 and the Satoyama will have any particular influence on each other, the Satoyama will always be a part of people’s lives. The way in which Satoyama is related to people’s lives will differ depending on the times.”

## 6. Conclusions

This study connects the results with four aspects of participants’ responses (within limitations). First, the use of green spaces in the city, which is important and valuable for the conservation and use of Satoyama, ought to be promoted. Second, regarding the participants of the experiment, young people may be encouraged to participate more in Sato-yama, increasing their opportunities to be close to nature, improving their physical and mental health, and relieving the stress caused by the COVID-19 pandemic. For the older adult with high blood pressure, maintaining Satoyama activity has a significant restorative effect on their hypertension; and it will allow older adult volunteers to see more value in their volunteer work. Third, in the post-COVID-19 era, in addition to the many popular parks and attractions, the physical and mental benefits of Satoyama activities are found in the small, unobtrusive green spaces around us, which are still easily accessible even in the city. Finally, the experimental method presented in this study was a challenge to other forest bathing health studies with precise timing and distance under completely human control. This study proved that this innovation can be used in a variety of activities.

## Figures and Tables

**Figure 1 ijerph-19-10760-f001:**
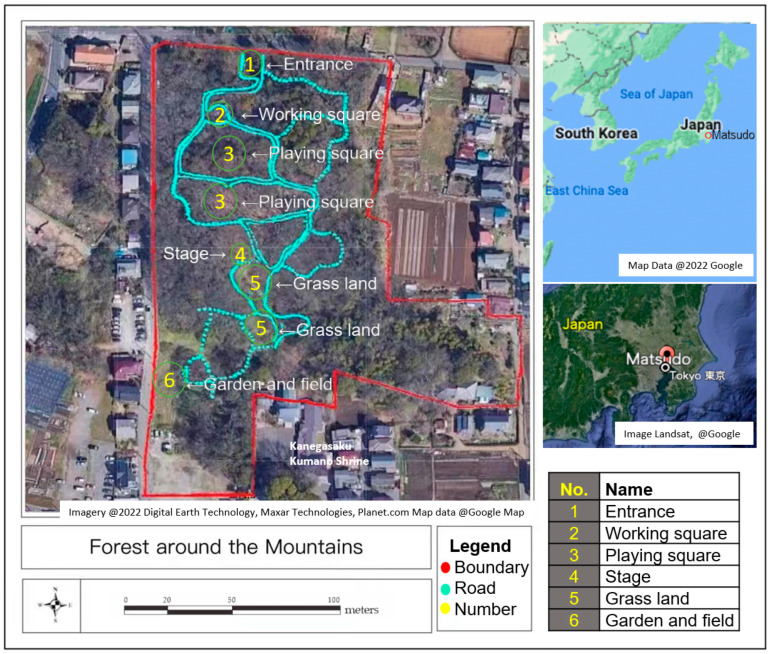
The location of the “Forest around the Mountains”. The red dotted lines were boundary line of the experimental site. The blue lines are the roads in the sample plot to delineate the specific area. The yellow numbers correspond to the names of these areas in English as explained in the lower right corner.

**Figure 2 ijerph-19-10760-f002:**
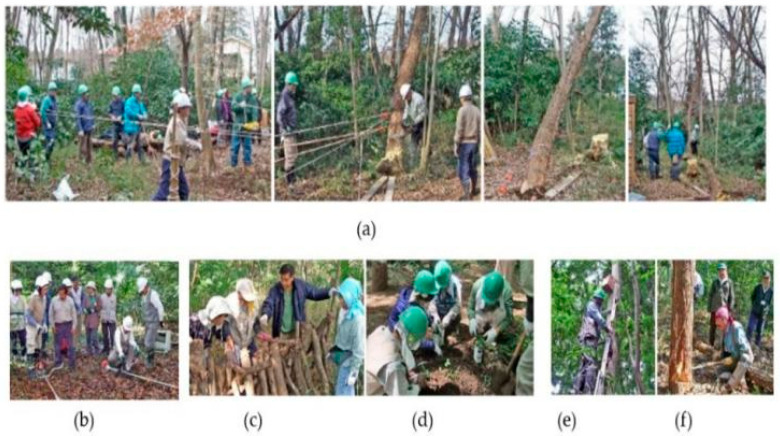
The usual activities in the “Forest around the Mountains”: (**a**) destroying a 70-year-old Robinia pseudoacacia; (**b**) using a mowing machine; (**c**) cultivating mushrooms; (**d**) cultivating Quercus serrata; (**e**) hanging a nest; (**f**) felling fir. The photo was taken by volunteers from the Forest around the Mountain NPO and provided by their homepage.

**Figure 3 ijerph-19-10760-f003:**
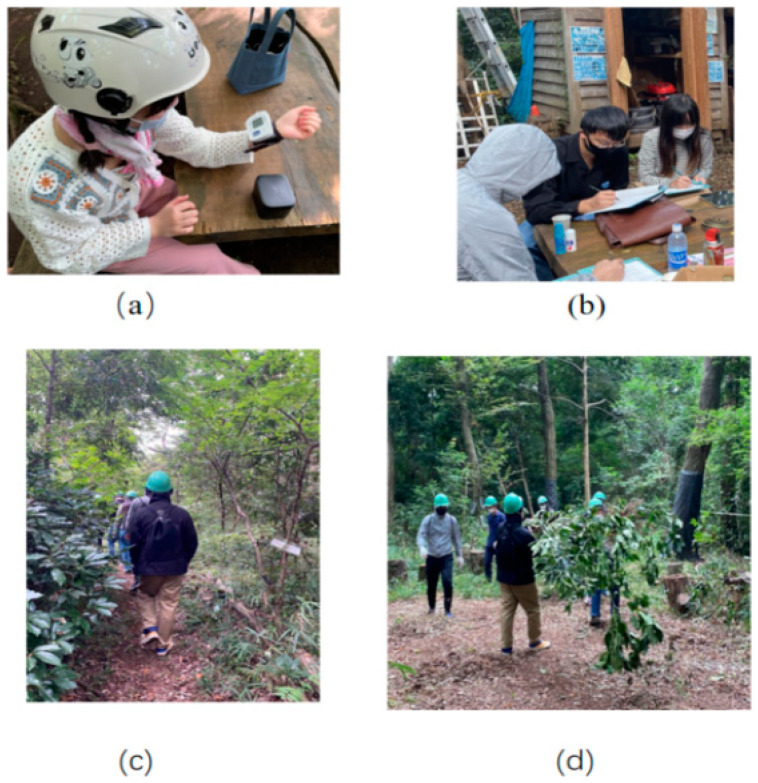
Experimental process: (**a**) physiological measurement; (**b**) answering the questionnaires; (**c**) nature observation; (**d**) Satoyama work.

**Figure 4 ijerph-19-10760-f004:**
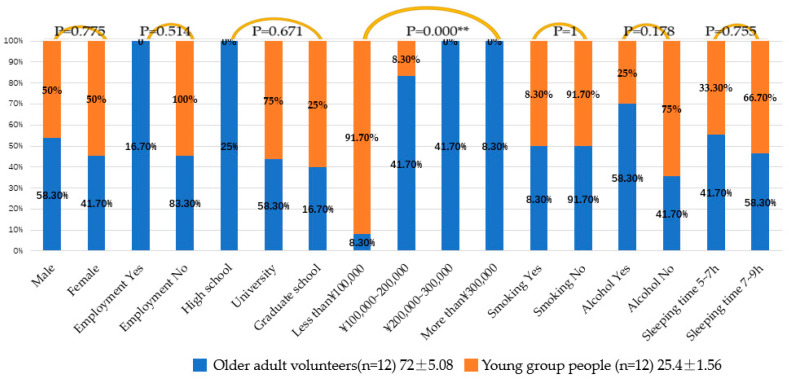
Participants’ information; the older adult volunteers (*n* = 12) and the group of younger people (*n* = 12). Note. ** *p* ≤ 0.01.

**Figure 5 ijerph-19-10760-f005:**
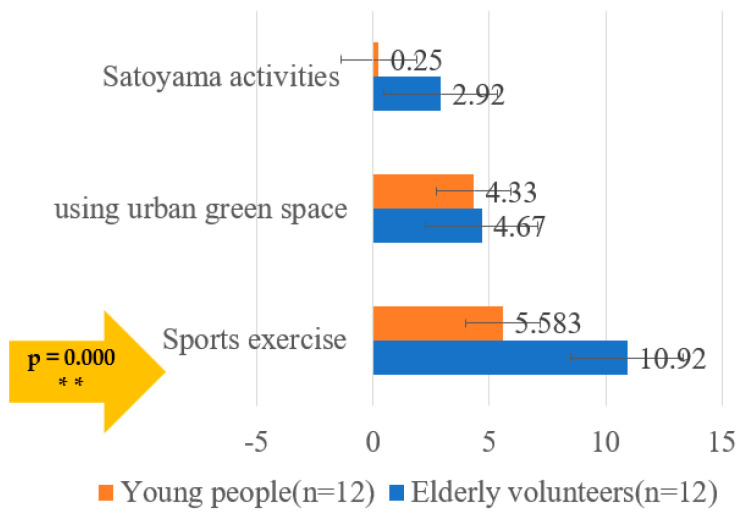
Proportion of participants’ habits of performing Satoyama activities and exercises and using urban green spaces monthly. Note. ** *p* ≤ 0.01.

**Figure 6 ijerph-19-10760-f006:**
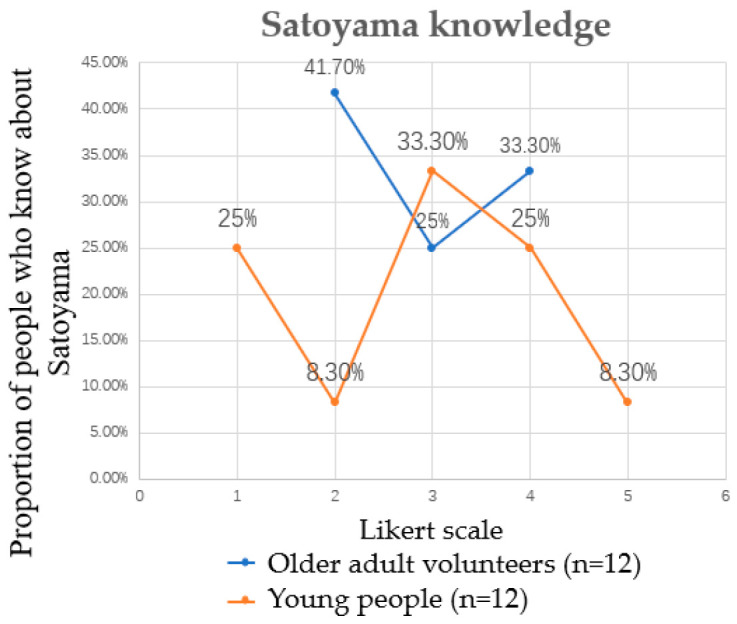
Proportion of people who know about Satoyama using a Likert scale. 1 = do not know at all; 2 = do not know much; 3 = neutral; 4 = know; 5 = know well.

**Table 1 ijerph-19-10760-t001:** The experiment flow in the “Forest around the Mountains”. The Profile of Mood States 2nd Edition is abbreviated as POMS. State-Trait Anxiety Inventory is abbreviated as STAI.

No	Time	Survey Flow
1	9:30–10 a.m.	Meet at the “Forest around the Mountains”
2	10:00–10:10 a.m.	Body temperature test and the disinfection of hands with alcohol
3	10:10–10:30 a.m.	Precautions for Satoyama activities and experiments
4	10:30–10:50 a.m.	Measuring blood pressure and heart rate and filling out POMS and STAI questionnaires before activity
5	10:50–11:20 a.m.	Nature observation (10 min), heart rate measurement
6	11:20–11:50 a.m.	Satoyama work (weeding, pruning branches, logging in 30 min), heart rate measurement
7	11:50–12 p.m.	Blood pressure, POMS, and STAI questionnaires measured after Satoyama activities
8	After 12 p.m.	Removal of the heart rate measuring device and finishing the experiment

**Table 2 ijerph-19-10760-t002:** The environmental physical indicators of 4 days in spring.

Parameters Mean ± SD	23-March	29-March	24-April	28-April
Temperature/°C	15.2 ± 2.66	23.5 ± 2.34	20.825 ± 1.08	22 ± 0.88
Relative humidity/%	54.25 ± 8.22	75.25 ± 6.95	39 ± 0.82	51.25 ± 1.71
Light/Klux	40.03 ± 10.60	26.77 ± 12.68	28.75 ± 22.27	24.67 ± 13.87
Sound/dB(A)	40.9 ± 1.3	44.5 ± 3.12	41 ± 1.41	47.33 ± 5.86

**Table 3 ijerph-19-10760-t003:** Cronbach’s α reliability analysis between older adult volunteers (*n* = 12) and young people (*n* = 12).

Cronbach’s α		
Parameter	Older Adult Volunteers (*n* = 12)	Young People (*n* = 12)
HR (RRI)	0.511	−1.660
Blood Pressure	0.684	0.863
POMS	0.767	0.795
STAI	0.806	0.680

**Table 4 ijerph-19-10760-t004:** Mean heart rate and blood pressure during Satoyama activity for older adult volunteers (*n* = 12) and young people (*n* = 12). Note. * *p* ≤ 0.05, ** *p* ≤ 0.01. The blue lines represent the *p*-values of the changes before and after Satoyama for the participants.

Physiological Parameters		Older Adults		Young People	*p*-Value
		Mean ± SD		Mean ± SD	
Systolic Blood Pressure					
Before Satoyama Activity	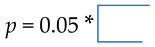	153.33 ± 26.46		114.58 ± 11.60	0.003 **
After Satoyama Activity		137.92 ± 16.88		113 ± 12.068	0.006 **
Diastolic Blood Pressure					
Before Satoyama Activity		91.58 ± 10.92	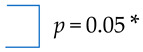	74.25 ± 13.66	0.023 *
After Satoyama Activity		84.08 ± 12.75		73.75 ± 14.04	0.084
Heart Rate					
Nature Observation		78.26 ± 2.89		96.28 ± 3.6	0.005 **
Satoyama Working		80.37 ± 4.18		81.65 ± 10.60	0.572

**Table 5 ijerph-19-10760-t005:** Mean of POMS and STAI in Satoyama activity between elder volunteers (*n* = 12) and young people (*n* = 12). POMS: Profile of Mood States; STAI: State-Trait Anxiety Inventory. Note. * *p* ≤ 0.05, ** *p* ≤ 0.01. The blue lines represent the *p*-values of the changes before and after Satoyama for the participants.

	Psychological Parameters		Older Adults		Young People		*p*-Value
			Mean ± SD		Mean ± SD		
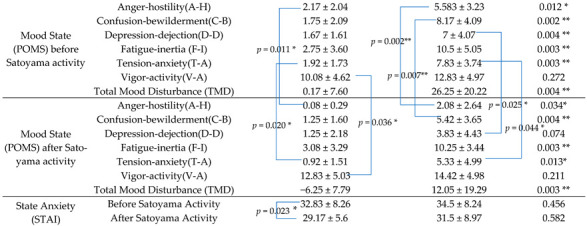							

## Data Availability

Data for this study is not publicly available.

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
