# Peer review of "Verification of Psychophysiological Effects of Satoyama Activities on Older Adult Volunteers and Young People in Post-COVID-19 Society: A Case Study of Matsudo City, Japan"

_ijerph, 2022, doi:10.3390/ijerph191710760_

Round 1
Reviewer 1 Report
I have uploaded my comments in a word file.

Author Response
Response to Reviewer 1 Comments
Point 1: My biggest criticism throughout the paper is a lack of rational explanation for what is being done. The author is very complete in explaining what was done but does not provide the reader with an understanding of why it was done. Why should the reader care about the topic in this paper? What value will this new knowledge have?
Response 1: Thank you very much for your comment and advice. Thank you very much for your comments, for which I have revised the purpose section such as L80 to L87 in the article: According to these researches, there are some questions as follows: (1) do small green areas in cities have restorative effects on people? (2) do Satoyama activities have restorative effects for different groups, especially older people who regularly volunteer in the forest and young people in school? It is assumed that it works better for young people who have no experience in Satoyama; (3) is it appropriate for people to conduct Satoyama activities during the COVID-19 pandemic, and do the experiments conducted in this period have any reference value for the recovery of people’s physical and psychological health?
Point 2: The author references Covid-19 as being important to this research in several places. However, the author never provides an adequate connection. How does this research relate to COVID-19? The connection either needs to be made clear or references to Covid-19 deleted.
Response 2: Thank you very much for your comment and advice. The link between this article and COVID-19 : In the post-COVID-19 era, many kinds of urban green space (UGS) are still proving to be beneficial to people’s physical and psychological health. In addition to the many popular parks and attractions, the physical and mental benefits of Satoyama activities are found in small, unobtrusive green spaces around us, which are still easily accessible even in the city. I added more literature related to COVID-19.
Point 3: The paper needs to be edited both for vocabulary and for punctuation. The author also needs to be consistent in the use of vocabulary. For example, the author refers to aims and also objectives. I would suggest that objective is the better word. The author also refers to subjects and two participants. Again, I would try to be consistent and would suggest using the word participants for this type of research.
Response 3: Thank you very much for your careful reading and detailed comments. These two points have been revised, and I will send the rest of the article to the line Editage and other professional English revision sites for checking.
Point 4: The abstract has too many acronyms. Many knowledgeable readers will not know all the acronyms. The author either needs to use the whole words or explain what's happening without abbreviations. I would suggest the latter
Response 4: Thank you very much for your suggestion. POMS and STAI changed into Profile of Mood States 2nd Edition and State-Trait Anxiety Inventor, A-H, D-D,T-A were changed into Anger-hostility, Depression-dejection, and Tension-anxiety values.
Point 5: Figure 1 should be placed in the text after it is referred to in the text. It doesn't seem to be in the right place
Response 5: Thank you very much for mentioning the situation. Figure 1 as our experimental location will appear as the initial location introduction, however in the experimental design Line122 and L131 need to shown where is the “working square” and “playing square” by the map, thus creating a sense of misalignment.
Point 6: The introduction should also tell the reader what's coming. The introduction should all tell the reader why this research is important or why the reader should care. It needs to be made explicit
Response 6: Thank you very much for your comments and suggestions. Something I changed in the introduction can explain the importance of this paper, from line 40 to line 46: governments’ policy of restricting outings and people’s fear of viruses may have caused some health problems during this period, including adverse psychological effects such as post-traumatic stress symptoms, confusion, anger, fear and discrimination… many kinds of urban green space (UGS) are still proving to be beneficial to people’s physical and psychological health [19-22]. many small areas of public green spaces have continued to offer a relaxing natural environment by using as Satoyama, although they are often neglected [23-25]. As the puepose shown (1) do small green areas in cities have restorative effects on people? (2) do Satoyama activities have restorative effects for different groups, especially older people who regularly volunteer in the forest and young people in school? It is assumed that it works better for young people who have no experience in Satoyama; (3) is it appropriate for people to conduct Satoyama activities during the COVID-19 pandemic, and do the experiments conducted in this period have any reference value for the recovery of people’s physical and psychological health?
Point 7: There needs to be a fuller explanation and discussion of Satoyama. There is quite a bit of research on the restorative aspects of nature. But, this almost seems contradictory to the work aspect of Satoyama. Most of the time we expect that work will cause more stress instead of stress relief. Could this be similar to an activity like gardening? There is some evidence that gardening is also a stress-relieving activity. If you take an evolutionarily perspective. The human mind takes a lot of time to develop and that is our evolution airy advantage. Therefore, humans have to " nurture the human mind, and perhaps that is why taking care of a garden or doing certain types of work is a restorative activity because they are nurturing. It may be something that feels natural and comfortable. I m not saying that this should be part of the article, but it seems to me that more discussion or even qualifying is needed
Response 7: Thank you very much for your comments. Work generally adds stress, even if it is gardening work, but the activities of the seniors in Satoyama are carried out as Matsudo Citizen Satoyama volunteers, and their usual participation in activities averages about three times a month, which is different from ordinary work, but instead makes their mental health very good, and their physical health can be seen to return to a normal level after the activities. This is reflected in the discussion section of the article (L331 to L335) [61, 62] According to a journal on exercise, heart rate goes from a rapid increase to a slow decrease after a period of exercise. Thus, as Satoyama works in our experiment were not highly difficult jobs but gardening works, such as weeding, pruning branches, and simple logging, we could not determine changes in heart rhythm for every minute [63].(L364 to L368)
Point 8: Figure 3 seems to have too much unnecessary detail about the equipment used. It destroys the flow of the paper I would suggest dropping figure 3.
Response 8: Thank you very much for your suggestion. Figure 3 was removed due to space being unnecessarily increased by the figures.
Point 9: I am concerned about the development of the variable " total emotional disturbance " it is a summation of a number of characteristics that were rated on a Likert scale. When you add them up it can increase the error and also it has no intuitive meaning. I know the author mentions that it has been used by others, but I really suggest that it doesn't make sense
Response 9: Thank you very much for the correction, I have removed the part describing "total emotional disturbance".
Point 10: I am also concerned about the reported heart rates during observation and during Satoyama work by young people. It seems like there is something wrong. It is counterintuitive that when engaging in work or soon after having engaged in work the participant's heartbeats would be lower and as reported in the results and quite a bit lower. If this is a correct result, so be it; but I think it deserves some explanation because it doesn't seem to make sense. Physical work raises heartbeats
Response 10: Thank you very much for your question. I will discuss this aspect in the limitation. Because working in the mountains is not a very difficult job: weeding and arranging branches. nature observation in the mountain is not just sitting in a certain position but walking from the entrance to the end of the forest, so from the first 10 minutes they were already in the activity, and when they work there is a rise in heart rate when they face the work, but also a gradual leveling off [64]. (L370 to L373)
Point 11: It seems contradictory in the abstract to say that so to Satoyama occurs in non-urban areas and then to follow it up shortly thereafter by saying Satoyama activities in small urban green spaces. Is it urban or not? It may just need some qualifying terminology
Response 11: Natural areas between cities and villages, such as grasslands, rice fields, drylands, and secondary forests, are widely defined as satoyama [26,27]. In Japan, rapid urban de-velopment in the 1970s forced some secondary forests used as fuel charcoal on the outskirts of cities to be left behind in the expanding cities [28]. By the 1990s the concept of Satoyama was expanded and these small green spaces could be used for new types of Satoyama activities: forest therapy [29], environmental education [30,31], and other places to conduct activities (concerts), besides forestry and agriculture [32]. (from line 58 to line 64 )
Point 12: The reporting of the results is complex because the author is looking at 2 different things. One is the pre and post-effects of Satoyama and they also look at differences between elderly people and younger people. A clearer explanation of the variables could help. What is the dependent variable and what are the independent variables? Again, what are the hypotheses or what is the new knowledge that the author is getting
Response 12: Thank you very much for your comments. I have updated Tables 4 and 5 in the hope that the reader will have a clearer idea of what we have done. The updated part compares the respective physiological and psychological data of the elderly and young people before and after the Satoyama activity, where the Satoyama activity is the independent variable and the physiological and psychological data are the dependent variables. The comparison between the elderly and the young was done with two equal independent samples, and the sample size was too small to satisfy the normal distribution and the t-test was dropped.
Point 13: The limitations section should acknowledge the limits of a relatively small sample size.
Response 13: Thank you very much for the correction. I add the same idea in limitation line 376 to line 378.
Point 14: Figure 7 does not seem to make sense. First, of all the RRI average was never really explained in the text, so the reader doesn't know what it is. Second, is supposed to represent both nature observation and Satoyama's work. How does it? How does it do that? Finally, I, m not sure what this figure tells us? Why is it important? I would suggest dropping it
Response 14: Thank you very much for your valuable comments. Figure 7. will be removed from the new article.
Reviewer 2 Report
Because of poor English writing, I stopped reading the manuscript after having read through the introduction section. Some of the grammatical mistakes are as follows: line 46 makes little sense; line 64 “are also have”; starting a paragraph with “however” in line 70; line 73 and lines 77-78, just to name a few. There are too many grammatical errors to hinder one’s understanding of the content. I recommend authors to have the manuscript proof-read by a professional English language service before submitting it for review.
Author Response
Response to Reviewer 2 Comments
Because of poor English writing, I stopped reading the manuscript after having read through the introduction section. Some of the grammatical mistakes are as follows: line 46 makes little sense; line 64 “are also have”; starting a paragraph with “however” in line 70; line 73 and lines 77-78, just to name a few. There are too many grammatical errors to hinder one’s understanding of the content. I recommend authors have the manuscript proofread by a professional English language service before submitting it for review.
I am sorry for my spelling mistake in the text, I checked the rest of the article after I corrected it here, as well as this article will be submitted to Editage (an agency that specializes in revising English essays) for a second time to check the English section.

Reviewer 3 Report
I think this is a timely and important report. There are a few remarks I would like to make and I would like to see them corrected.
Title: This paper is a case study of activities in satoyama, Japan. The title should include the word "Japan".
Also, COVID-19 was and is an ongoing issue when this study was conducted. The phrase "post-COVID 19 Society" should be changed to "during" or some other accurate time expression.
L.39-42: The context before and after here reads like a description of the impact of COVID-19, but the references cited are pre-epidemic. I think you need to separate the general discussion from the impact under the COVID-19 epidemic.
L.83: Please add the latitude and longitude of the Study site.
L.108-109: Whether the young people originally had experience or interest in satoyama conservation activities may affect the results of this study. Please describe whether they were interested in satoyama activities before this experiment.
L.190-192: Since the young people are either college or graduate students, was their last education in high school or college, respectively? If so, please revise the text description and recalculate the statistical analysis, as there might be a significant difference.
L.210: Satiyama? -> Satoyama
Fig.5: Please change the color of the bar indicating youth and elderly to the same color as in Fig.4 and 6.
L.280-294: I don't understand the necessity of this paragraph in the discussion. What is written here is the study design and I believe should be described in the Methods.
The first sentence describes the situation in Japan, but no evidence information is given. Even within the same country, the situation may differ from city to city.
L.293: he 30 minutes of work ?
Author Response
Response to Reviewer 3 Comments
I think this is a timely and important report. There are a few remarks I would like to make and I would like to see them corrected.
Title: This paper is a case study of activities in satoyama, Japan. The title should include the word "Japan".
Also, COVID-19 was and is an ongoing issue when this study was conducted. The phrase "post-COVID 19 Society" should be changed to "during" or some other accurate time expression.
Response 1: Thank you very much. Based on your valuable comments, I changed the title of the article with a more accurate meaning.
L.39-42: The context before and after here reads like a description of the impact of COVID-19, but the references cited are pre-epidemic. I think you need to separate the general discussion from the impact under the COVID-19 epidemic.
Response 2: Thank you very much for your comment and advice. Based on your valuable comments, I have made the following changes in the line 40 to 46:As proven by researchers, governments’ policy of restricting outings and people’s fear of viruses may have caused some health problems during this period, including adverse psychological effects such as post-traumatic stress symptoms, confusion, anger, fear and discrimination, anxiety, depression, and stress about the sustainability of the current job [9-12]. This socially destructive stress can cause serious problems related to the proper functioning of the immune system [13].
L.83: Please add the latitude and longitude of the Study site.
Response 3: Thank you very much for your valuable comments. I added the description of latitude and longitude in lines 92 and 93.
L.108-109: Whether the young people originally had experience or interest in satoyama conservation activities may affect the results of this study. Please describe whether they were interested in satoyama activities before this experiment.
Response 4: Thank you very much for your comments The questionnaire for personal information was filled out before conducting the experiment, and the question asked about their level of knowledge about Satoyama and the level of participation in Satoyama, as can be seen in Figures 5 and 6.
L.190-192: Since the young people are either college or graduate students, was their last education in high school or college, respectively? If so, please revise the text description and recalculate the statistical analysis, as there might be a significant difference.
Response 5: Thank you very much for your valuable comments. There is no high school for final education. College students and graduate school graduates are their final education.
L.210: Satiyama? -> Satoyama
Response 6: Thank you very much for your valuable comments. I am very sorry for my spelling mistakes in the text, I checked the rest of the article after I corrected it here, as well as this article will be submitted to Editage for a second time for checking the English part.
Fig.5: Please change the color of the bar indicating youth and elderly to the same color as in Fig.4 and 6.
Response 7: Thank you very much for your comments. I have modified the color bar of the elderly in Figure 5 to be consistent with Figure 4 and Figure 6.
L.280-294: I don't understand the necessity of this paragraph in the discussion. What is written here is the study design which I believe should be described in the Methods.
The first sentence describes the situation in Japan, but no evidence information is given. Even within the same country, the situation may differ from city to city.
Response 8: Thank you very much for your correction. This is one of the features that distinguish this experiment from others because the normal activities invested in Riyama are not just walking and sitting and watching under very strict control, so I have placed this fragment in the experimental design as a support for the experimental basis.
L.293: he 30 minutes of work
Response 9: Thank you for your valuable correction. The “he” of this sentence is wrong, “the” is right.

Reviewer 4 Report
Dear Authors, I carefully read your manuscript and I think it should be generally improved. A major concern is related to the present form. I strongly suggest reducing the lenght of the manuscript and focusing on the most relevant findings being more concise and precise. The readers should appreciate the great effort you have done and you could help by enhancing the general form (especially methods description and data presentation) of your manuscript. Another major concern is about the statistical test chosen. Some of them should be reconsidered (or their choice supported by additional tests, please see my following comments).
Abstract
I would suggest adding some quantitative data in the results section to supply the readers’ comprehension on the impact of the forest bath on psychological and physiological aspects. Please remember that The abstract should be a total of 200 words maximum.
Main text
Paragraph 2.3 Experimental Design: please try to order the phases described in this paragraph following a more defined workflow (as you did in the table 1). In the present form, the paragraph is quite difficult to follow.
Line 119: please define POMS and STAI here. It is the first time you mention them in the main text. Also, add details about the instruments used to measure blood pressure;
Line 127: please define NPO for the first time here;
Lines 171-73: I would strongly suggest running a non-parametric test instead (and amend the results accordingly). The sample size is very small and this suggest there could be the violation of the assumptions behind the t-test. Otherwise, please provide that data are normally distributed (in both groups) and variance is homogenous (in both groups).
Table 1: please add all relevant acronym explanation to let anyone to read the table as a stand-alone element without necessarily referring to the main text. Please do the same for all the other tables/figures in the text.
Paragraph 3. Results: in general this paragraph should be improved. Starting from the “3.1. Environment, Participants and Satoyama activity Information” where a lot of provided information are not previously described in the material and methods section. All the results provided should have a correspondence in the material and methods paragragh (e.g. humidity and temperature have been measured by the mentioned LCD mentioned in lines 138-139? This is not totally clear and follows a bit confusing scheme. I strongly suggest maintaining also the same order in presenting the description of material and methods and results (e.g. first subject’s characteristics in both, then environmental parameters in both, etc.).
Table 4: please report the name of the statistical test used
Paragraph 5. Limitations: please add also the strengths of your work.
Paragrapf 6. Conclusions: please do not repeat the results here. I strongly suggest focusing more on the impact that your findings could have.
Author Response
Response to Reviewer 4 Comments
Dear Authors, I carefully read your manuscript and I think it should be generally improved. A major concern is related to the present form. I strongly suggest reducing the lenght of the manuscript and focusing on the most relevant findings being more concise and precise. The readers should appreciate the great effort you have done and you could help by enhancing the general form (especially methods description and data presentation) of your manuscript. Another major concern is about the statistical test chosen. Some of them should be reconsidered (or their choice supported by additional tests, please see my following comments).
Response: Thank you for your valuable comments and advice. According to your request, I made 10 replies and made profound modifications to my article, especially in the analysis method, and obtained more reliable results.
Abstract
I would suggest adding some quantitative data in the results section to supply the readers’ comprehension on the impact of the forest bath on psychological and physiological aspects. Please remember that. The abstract should be a total of 200 words maximum.
Response 1: Thank you very much for your comments. I shortened the abstract to about 250 words.
Main text
Paragraph 2.3 Experimental Design: please try to order the phases described in this paragraph following a more defined workflow (as you did in the table 1). In the present form, the paragraph is quite difficult to follow.
Response 2: Thank you very much for your valuable comments. The Line 122 to Line 138 sections conform to the order in Table 1 and are more convenient for the reader to read
Line 119: please define POMS and STAI here. It is the first time you mention them in the main text. Also, add details about the instruments used to measure blood pressure;
Response 3: Thank you very much for your careful reading and valuable comments, I have added the full POMS and STAI write-ups here along with abbreviations. The section on the use of sphygmomanometers is also added in 2.4. Experimental equipment and questionnaire.
Line 127: please define NPO for the first time here;
Response 4: Thank you very much for pointing this out, I have added a detailed writeup of the NPO in 2.2. Study Subject Line112 and the reference [49].
Lines 171-73: I would strongly suggest running a non-parametric test instead (and amend the results accordingly). The sample size is very small and this suggest there could be the violation of the assumptions behind the t-test. Otherwise, please provide that data are normally distributed (in both groups) and variance is homogenous (in both groups).
Response 5: Thank you very much for your correction, the sample size was tested to be not normally distributed, so like the psychological data, the Wilcoxon signed rank test was also used for the physiological data as a non-parametric test. And the p-values of the two comparison samples were modified in Table 4.
Table 1: please add all relevant acronym explanation to let anyone to read the table as a stand-alone element without necessarily referring to the main text. Please do the same for all the other tables/figures in the text.
Response 6: Thank you very much for the correction. I have explained the original meaning of the abbreviated letters POMS,STAI in the section of Table 1 description.
Paragraph 3. Results: in general this paragraph should be improved. Starting from the “3.1. Environment, Participants and Satoyama activity Information” where a lot of provided information are not previously described in the material and methods section. All the results provided should have a correspondence in the material and methods paragragh (e.g. humidity and temperature have been measured by the mentioned LCD mentioned in lines 138-139? This is not totally clear and follows a bit confusing scheme. I strongly suggest maintaining also the same order in presenting the description of material and methods and results (e.g. first subject’s characteristics in both, then environmental parameters in both, etc.)
Response 7: Thank you very much for your careful reading and valuable comments. I added that the environment was measured with an LCD digital wireless indoor / outdoor Thermometer Hygrometer. (line 201 to 202) Since I mentioned the experimental address first in Materials and Methods, the results should also be consistent in expressing the environment of the experimental address first, and since moving the graphs would bring an unnecessary increase in space, the structure of this section has been retained.
Table 4: please report the name of the statistical test used
Response 8: Thank you very much for your valuable comments. I added the mean pairwise comparisons by the nonparametric test--Wilcoxon Signed Ranks Test for testing p-values in lines 194 and 198.
Paragraph 5. Limitations: please add also the strengths of your work.
Response 9: Thank you very much for your suggestion, but I have modified the strong points of this study in disscusion
Paragrapf 6. Conclusions: please do not repeat the results here. I strongly suggest focusing more on the impact that your findings could have.
Response 10: Thank you very much for your suggestion, I revised the conclusion section: This study connects the results with four aspects of the participant's responses (in the limitation). First, the use of green spaces and lakes in the city, which is important and valuable for the conservation and use of Satoyama, ought to be promoted. Second, with regard to the participants of the experiment, young people may be encouraged to participate more in Satoyama, increasing their opportunity to get close to nature, improving their physical and mental health, and relieving the stress caused by the COVID-19 pandemic. For the older adult, maintaining Satoyama activity will allow them to see their volunteer work's value. Third, in the post-COVID-19 era, in addition to the many popular parks and attractions, the physical and mental benefits of Satoyama activities are found in small, unobtrusive green spaces around us, which are still easily accessible even in the city. Finally, the experimental method presented in this study was a challenge to other forest bathing health studies with precise timing and distance under completely human control. This study proved that this innovation can be used in a variety of activities.

Round 2
Reviewer 1 Report
Nice work. I am anxious to see where your research goes from here.
Author Response
Thank you very much for your comments and support!I have learned a lot and improved a lot after you pointed out the shortcomings in my article.
Reviewer 2 Report
Overall comments
Much improved. After making some necessary revisions as below, the paper can be accepted for publication.
Specific comments
Title
Suggest using “in” instead of “during.”
Line 216
Is income a monthly or an annual income?
Lines 253-255
In the discussion section, please explain what these numbers mean.
Line 297
Didn’t V-A increase after Satoyama activities? Please check and correct.
Line 298
“No significant difference”? In Table 5, C-B and F-I have **. Please check.
Line 324-350
Tables 4 and 5 should be referred to in the text where relevant.
Figure 4
This figure must be revised. The upper half illustration is not a standard representation of the information.
Caption: correct “adultselderly.”
Table 3
Please correct “a” to alpha.
Table 4
Please adjust the location of the arrow (p=0.05).
Author Response
Response to Reviewer 2 Comments
Overall comments
Much improved. After making some necessary revisions as below, the paper can be accepted for publication.
Response: Thank you very much for your time and patience in reading, and responses 1 to 10 of the paper.
Specific comments
Title
Suggest using “in” instead of “during.”
Response 1: Thank you very much. Based on your valuable comments, I changed “in” instead of “during”.
Line 216
Is income a monthly or an annual income?
Response 2: Thank you very much for your comments. The income belongs to monthly income. I modified it "Income of per month was measured in Japanese yen." from line 214 to line 215.
Lines 253-255
In the discussion section, please explain what these numbers mean.
Response 3: Thank you very much for your comments, I have modified the 251 to 255 part of the result: Cronbach’s α reliability analysis between the older adult volunteers (n = 12) and young people (n = 12) showed that the reliability of the POMS score for the two groups was 0.965 and 0.824 with excellent internal consistency, respectively; and that of the STAI score was 0.841 and 0.681 with good internal consistency, respectively [60].
Line 297
Didn’t V-A increase after Satoyama activities? Please check and correct.
Response 4: Thank you for your correction. V-A should not appear in the description of line 297. Because it doesn’t belong to the negative emotion value and decreased a little after Satoyama activities: it was 0.272 before activities and 0.211 after activities without significance, I canceled the description of this value.
Line 298
“No significant difference”? In Table 5, C-B and F-I have **. Please check.
Response 5: Thank you for your comments. C-B and F-I have significant differences and are marked in line 301 which was the same in table 5.
Line 324-350
Tables 4 and 5 should be referred to in the text where relevant.
Response 6: Thank you for your valuable advice. I marked table4 in line 325. Table5 appears on lines 337, 343, 346, and 347.
Figure 4
This figure must be revised. The upper half illustration is not a standard representation of the information.
Response 7: Thank you very much for your comments. Due to space constraints in the article, this type of bar chart had to be used. But I have added data labels. And the p-value has more description to avoid misunderstanding in the paper. It represents the significant difference in each indicator between the two groups of older adult volunteers and young people. From line 204 to line 225 is the description around Figure 4.
Caption: correct “adults elderly.”
Response 8: Thank you very much for your correction. It has been changed to " Figure 4. Participants information of the older adult volunteers (n=12) and young group people (n=12). "
Table 3
Please correct “a” to alpha.
Response 9: Thank you very much for your comments. “Cronbach's a” has been changed to “Cronbach's α” in table 3.
Table 4
Please adjust the location of the arrow (p=0.05).
Response 10: Thank you very much for your correction. location of the arrow has been adjusted between 153.33±26.46 and 137.92±16.88; 91.58±10.92 and 84.08±12.75.
